# Degradation Kinetics and Shelf Life of *N*-acetylneuraminic Acid at Different pH Values

**DOI:** 10.3390/molecules25215141

**Published:** 2020-11-05

**Authors:** Weiwei Zhu, Xiangsong Chen, Lixia Yuan, Jinyong Wu, Jianming Yao

**Affiliations:** 1Institute of Plasma Physics, Hefei Institutes of Physical Science, Chinese Academy of Sciences, Hefei 230031, China; zw840602@mail.ustc.edu.cn (W.Z.); xschen@ipp.ac.cn (X.C.); jywu@ipp.ac.cn (J.W.); 2University of Science and Technology of China, Hefei 230026, China

**Keywords:** degradation kinetics, pH dependence, hydrogen peroxide, shelf life, *N*-acetylneuraminic acid

## Abstract

The objective of this study was to investigate the stability and degradation kinetics of *N*-acetylneuraminic acid (Neu5Ac). The pH of the solution strongly influenced the stability of Neu5Ac, which was more stable at neutral pH and low temperatures. Here, we provide detailed information on the degradation kinetics of Neu5Ac at different pH values (1.0, 2.0, 11.0 and 12.0) and temperatures (60, 70, 80 and 90 °C). The study of the degradation of Neu5Ac under strongly acidic conditions (pH 1.0–2.0) is highly pertinent for the hydrolysis of polysialic acid. The degradation kinetics of alkaline deacetylation were also studied. Neu5Ac was highly stable at pH 3.0–10.0, even at high temperature, but the addition of H_2_O_2_ greatly reduced its stability at pH 5.0, 7.0 and 9.0. Although Neu5Ac has a number of applications in products of everyday life, there are no reports of rigorous shelf-life studies. This research provides kinetic data that can be used to predict product shelf lives at different temperatures and pH values.

## 1. Introduction

Sialic acids comprise a family of over 80 nine-carbon acidic monosaccharides, which are *N*- or *O*-substituted derivatives of neuraminic acid, with *N*-acetylneuraminic acid (Neu5Ac) as the most abundant representative [1]. The molecular structure of Neu5Ac has been characterized and is documented in the literature [2,3,4,5]. Neu5Ac is a natural monosaccharide that is widely distributed in higher animals, with the highest concentrations detected in brain tissue and milk [6]. Neu5Ac occupies the terminal position of membrane glycoproteins, and is involved in many biological phenomena such as glycoprotein stability, neuronal sprouting and plasticity, cell adhesion, trapping of pathogens and cell signaling [7,8,9,10].

Neu5Ac has a high potential for use as a novel food additive due to its attractive biological effects. In January 2016, the Food and Drug Administration of the United States announced the submission of Neu5Ac safety assessment materials by Glycom of Denmark. The product is considered safe as a food raw material and can be used in full-term infant formula, traditional foods and beverages. In May 2017, the National Health Commission of the People’s Republic of China examined and approved the safety assessment materials of Neu5Ac. In December 2017, the European Food Safety Authority launched Neu5Ac as a novel food ingredient.

In the course of our tests we found that the degradation rate of Neu5Ac is highly dependent on the storage conditions. It has been reported that the stability and shelf life of processed food depend on several factors, including pH, temperature, heating time and oxidation [11,12,13,14,15,16,17,18,19,20]. It is therefore important to study the effects of pH, temperature heating time and oxidation on Neu5Ac.

The aim of this study was to analyze the influence of pH and temperature on the stability of Neu5Ac in aqueous solution. Autoclaving was used as the primary screening method for selecting the pH values at which Neu5Ac is stable even at high temperatures. Subsequently, the degradation kinetics of Neu5Ac at unsuitable pH values at different temperatures were analyzed. Then, the effect of H_2_O_2_ on the stability of Neu5Ac at three pH values (pH 5.0, 7.0 and 9.0) was studied. Finally, we estimated the shelf life and shelf life variability of Neu5Ac when stored at selected pH values and temperatures. There are few reports on the determinants of the storage stability of Neu5Ac. The objective was to study the stability of Neu5Ac at various pH values, temperatures and under oxidative conditions in order to determine the stability of Neu5Ac during processing, food production, distribution and storage. Our data provide a theoretical guidance for the future research on the practical utilization of Neu5Ac.

## 2. Results and Discussion

### 2.1. Impact of Autoclaving on the Stability of Neu5Ac

To determine stability at twelve pH values (1.0, 2.0, 3.0, 4.0, 5.0, 6.0, 7.0, 8.0, 9.0, 10.0, 11.0 and 12.0), Neu5Ac solution were autoclaved at 121 °C for 20 minutes. Figure 1 shows the effects of pH on the thermal stability of Neu5Ac. The results indicate that Neu5Ac is comparably stable at pH 3.0–10.0. Especially at pH 7.0, more than 99.0% of the initial Neu5Ac concentration remained, even with heating at 121 °C for 20 min.

However, the remaining amount of Neu5Ac decreased significantly at pH 1.0, 2.0, 11.0 and 12.0. At pH values of 1.0, 2.0, 11.0 and 12.0, only 14.0%, 25.0%, 13.0% and 11.0% of the Neu5Ac remained after heating at 121 °C for 20 min, respectively. Therefore, Neu5Ac should be more heat-stable when it is used in food products with a neutral pH, while too high or too low pH hastened the degradation of Neu5Ac. Therefore, a detailed study of the thermal stability of Neu5Ac in relation to pH was conducted in three pH ranges: pH 1.0–2.0, pH 3.0–10.0 and pH 11.0–12.0.

Autoclaving was used as the primary screening method for selecting the heat-stable pH values at high temperatures. The thermal degradation of Neu5Ac at pH 3.0–10.0 was limited whereas significantly at pH 1.0, 2.0, 11.0 and 12.0. Neu5Ac showed good thermal stability at pH 3.0–10.0, and this pH range was also commonly used in food processing, which is of guiding significance for the processing of foods containing Neu5Ac.

Neu5Ac shows strong acidity after dissolving in water. (2% aqueous solution: pH 1.8–2.3). Adding it directly to the aqueous solution of products such as beverages will have an impact on the pH value of the products. The results of our research could guide the determination of the best range of the final pH value of the product.

Hydrolysis of polysialic acids, which is carried out under strongly acid conditions, is one of the main preparation methods for Neu5Ac [21]. Therefore, characterizing the degradation kinetics of Neu5Ac at pH 1.0–2.0 is highly pertinent to its industrial application. Decarboxylation is a fundamental and important reaction in organic synthesis and drug discovery, which is generally catalyzed by acids or bases [22]. Furthermore, alkaline deacetylation is a common synthetic method in chemistry [23]. We assumed that the decarboxylation and deacetylation of Neu5Ac readily occurs with heating at low pH values. At high pH values, alkaline deacetylation occurs.

### 2.2. Stability of Neu5Ac during Heat Treatment at Different pH Values

Neu5Ac was quite stable during heat treatment at pH 3.0–10.0, so we only studied the stability of Neu5Ac at four extreme pH values: 1.0, 2.0, 11.0 and 12.0. The effects of pH on the heat stability of Neu5Ac are presented in Figure 2. The results indicated that Neu5Ac was more stable at low temperatures at all the tested pH values. After heating at 60 °C for 6 h, 91.5%, 94.5%, 88.1% and 45.1% of the initial Neu5Ac remained at pH 1.0, 2.0, 11.0 and 12.0, respectively. Furthermore, the remaining content decreased as the temperature increased, and after heating for 6 h at 90 °C, only 48.0%, 59.6%, 36.0% and 1.5% of the initial Neu5Ac remained. Notably, Neu5Ac was more stable when exposed to strong acid (pH 1.0 and pH 2.0) than to strong alkali (pH 11.0 and pH 12.0). At pH 1.0, 48.0% of the initial Neu5Ac content remained after heating at 90 °C for 6 h. By contrast, only 45.1% of the initial content remained even at the lowest tested temperature of 60 °C when the pH was increased to 12.0. After heating for 6 h at 90 °C only 1.5% of the Neu5Ac remained.

### 2.3. Kinetics of the Thermal Degradation of Neu5Ac

The kinetics of the thermal degradation of Neu5Ac were analyzed using Equation (1). Higher temperatures and longer times increased the degradation (Figure 2). First-order reaction kinetics could describe the thermal degradation of Neu5Ac at four pH values (1.0, 2.0, 11.0 and 12.0). The rate constant (k) and regression coefficient (R^2^) for the degradation of Neu5Ac at different temperatures were then calculated from the slope of ln(C/C_0_) versus time, using linear regression analysis for first-order reaction kinetics. First-order kinetics described the thermal degradation of Neu5Ac well. Under the given conditions, the decrease of Neu5Ac concentration fitted a first-order equation with a good regression coefficient (0.9422 < R^2^ < 0.9959).

The Arrhenius relationship of the rate constants (ln k) was plotted against the reciprocal of temperature (1/T) at various pH values as shown in Equation (4). The slopes at pH 1.0, 2.0, 11.0 and 12.0 were −8569.3, −8720.5, −8280.6 and −7036.9, respectively. The corresponding R^2^ values were satisfactory, at 0.9943, 0.9901, 0.9891 and 0.9903, respectively (Figure 3). The results show that the ln k was linear with the 1/T in the temperature range of 60–90 °C, and k increased with rising T at selected pH values. The degradation of Neu5Ac therefore accelerated by the increase of temperature.

Table 1 shows the kinetic parameters obtained for each evaluated temperature. The kinetic rate constants (k) increased with increasing temperature. The k indicator enables the prediction of the thermal degradation rate. The lower the k value for a certain set of conditions, the better the stability Neu5Ac under said conditions. The k values significantly increased as the temperature increased from 60 to 90 °C at all four pH values. The decimal reduction (D-values) and the half-time (t_1/2_) values for each experiment decreased as the incubation temperature increased. The D-values and t_1/2_ decreased sharply under strongly alkaline conditions compared with strong acid at the same temperature. For example, the t_1/2_ value of Neu5Ac degradation decreased significantly from 72.96 to 0.93 h as the temperature and pH increased from 60, pH 2.0, to 90 °C, pH 12.0. Furthermore, the activation energy (Ea) decreased from 72.50 to 58.50 kJ/mol as the pH increased from pH 2.0 to pH 12.0. The Ea was the largest and the Z-value was the smallest at pH 2.0, indicating that a smaller temperature change was needed to degrade the compound more rapidly. Therefore, Neu5Ac was less susceptible to thermal degradation at pH 2.0 than at the other three tested pH values.

The ΔG value reflects the spontaneous direction of a reaction under the given conditions. A positive sign indicated that degradation is a nonspontaneous reaction. At each pH value, the ΔH values were similar for all heating conditions. This indicated that the energy barrier that must be overcome in order to achieve the transition was similar. The positive sign of ΔH means that the degradation is an endothermic reaction. ΔS measures the change of disorder among the molecules in the system. At a given pH value, ΔS was less affected by the temperature. The absolute value of ΔS was 95.55–97.18 J/mol at pH 12.0, which was significantly higher than the value of 74.19–76.14 J/mol at pH 2.0. This indicated that Neu5Ac was practically not sensitive to the tested temperature at pH 2.0. The stability of Neu5Ac at the four pH values was in the order pH 2.0 > pH 1.0 > pH 11.0 > pH 12.0.

The production process of Neu5Ac approved by the National Health Commission of the people’s Republic of China is: food-grade glucose and corn pulp as raw materials, *Escherichia coli* (strain number SA-8) fermentation, filtration, sterilization, hydrolysis, purification and other processes. The hydrolysis process needs to hydrolyze polysialic acid to prepare Neu5Ac, under the condition of strong acid and high temperature, so the study of degradation kinetics under strong acid can guide the production.

In order to do a comprehensive study on the influence of pH during heat treatment of Neu5Ac, we also studied the effect of strong alkali. Alkaline deacetylation is also a common synthetic method in chemistry. The thermal degradation of Neu5Ac followed first-order reaction kinetics at all investigated temperatures. The degradation under the influence of strong acid and strong alkali was nonspontaneous and endothermic.

### 2.4. The Effect of H_2_O_2_ on the Stability of Neu5Ac

The thermal degradation of Neu5Ac at pH 3.0–10.0 was limited, which emphasizes the importance of oxidants, exemplified by H_2_O_2_ under these conditions. We therefore investigated effect of H_2_O_2_ on the stability of Neu5Ac at pH 5.0, 7.0 and 9.0. The results revealed that the addition of H_2_O_2_ reduced the stability of Neu5Ac in a concentration-dependent manner. Even though Neu5Ac was stable at pH 3.0–10.0, even with heating, the addition of H_2_O_2_ had a greater impact on its stability. The plots of ln (C_t_/C_0_) versus time at various concentrations of H_2_O_2_ are shown in Figure 4. The degradation rate constants (k) in the presence of 4% H_2_O_2_ at pH 5.0, 7.0 and 9.0 were 0.3255, 0.3095 and 0.2819, respectively. The pH stability of Neu5Ac in the presence of 4% H_2_O_2_ was in the order pH 9.0 > pH 7.0 > pH 5.0. The high values of R^2^ (R^2^ > 98.2%) under all conditions revealed that the process of Neu5Ac oxidation by H_2_O_2_ was consistent with first-order kinetics.

H_2_O_2_ can generate highly active free radicals such as ·OH and ·O_2_^−^, which are the key species responsible for the oxidative degradation of organic matter. We assumed that the free radicals could directly degrade Neu5Ac in aqueous solution, as discussed earlier [16]. Notably, the amount of ·OH formed in alkaline solutions is known to be lower than in acidic solutions [24]. Hence, the concentration of ·OH was very low in alkaline solutions, so the degradation of Neu5Ac was weaker than in acidic solutions. If exposed to an oxidizer for a short time, the alkaline condition will be more stable. Although the degradation rate was lowest at high pH values, the k constant was less affected by the pH. The addition of oxidants such as H_2_O_2_ to Neu5Ac should be avoided, and it should be stored in an anoxic environment.

### 2.5. Determination of the Shelf Life of Neu5Ac in Aqueous Solution at 25 and 50 °C

The shelf life at 25 and 50 °C was determined at all three studied intermediate pH values. As shown in Figure 5, the decrease of the Neu5Ac concentration fitted a first-order equation with a good regression coefficient (0.9434 < R^2^ < 0.9994). When viewing the storage-related decay as a first-order reaction, the shelf life can be determined by Equation (8). Shelf life was defined as the time required to reach 90% of the labeled amount under defined storage conditions [25]. The results showed that the potential shelf life of Neu5Ac at pH 5.0, 7.0 and 9.0 at the reference temperatures of 25 °C was 199, 817 and 387 days, while it was 47, 104 and 77 days at 50 °C, respectively. The results clearly indicated that Neu5Ac would have an estimated shelf life of at least 2 years when formulated at its optimum pH and stored at room temperature (25 °C). On the other hand, the shelf life at pH 5.0 and 25 °C was calculated to be 199 days. We can infer that specific H^+^ catalysis occurred at pH 5.0, and OH^−^ specific degradation was present at pH 9.0. The optimum pH for stability was 7.0, with the lowest relative amounts of OH^−^ and H^+^ [25]. It is important to select appropriate storage conditions in terms of pH and temperature depending on the intended use of Neu5Ac.

## 3. Materials and Methods

### 3.1. Reagents

Neu5Ac with a purity of >98% (HPLC) was kindly provided by Wuhan Zhongke Optics Valley Green Biotechnology Co., Ltd. (Wuhan, China). Distilled water was used throughout the study. All other reagents were of analytical grade and were purchased from Sinopharm Chemical Reagent Co., Ltd. (Shanghai, China). 

### 3.2. Preparation of the Neu5Ac Solution

Neu5Ac (about 2000 mg) was dissolved in 200 mL of distilled water, and the pH was adjusted with 2 mol/L NaOH and 2 mol/L HCl where appropriate. Aliquots comprising 10 mL of the Neu5Ac solution were distributed into three glass bottles hermetically sealed with aluminum lids.

### 3.3. Sample Preparation for Autoclaving

The thermal stability of Neu5Ac at twelve pH values (1.0, 2.0, 3.0, 4.0, 5.0, 6.0, 7.0, 8.0, 9.0, 10.0, 11.0 and 12.0) was studied by autoclaving. The sample bottles (3 bottles per each pH) were autoclaved at 121 °C for 20 min. After the heat treatment, 1 mL samples of the Neu5Ac solution were pipetted from each bottle and filtrated through syringe filters (MCE, diameter 13 mm, pore size 0.22 µm, Navigator Lab Instrument Co., Ltd.). The filtrate was removed, rapidly cooled on ice to quench the reaction, and analyzed by high-performance liquid chromatography (HPLC). 

### 3.4. Sample Preparation for the Measurement of the Degradation Kinetics and Thermodynamic Studies

The effect of pH on the thermal stability of Neu5Ac was studied at four pH values (1.0, 2.0, 11.0 and 12.0) and four temperatures (60, 70, 80 and 90 °C). The sample bottles (3 bottles per each pH) were placed in a thermostatic water bath preheated to a given temperature (60, 70, 80 and 90 °C). During the heating process, 1 mL samples of the Neu5Ac solution were pipetted from each bottle every 1 h and filtrated through syringe filters (MCE, diameter 13 mm, pore size 0.22 µm, Navigator Lab Instrument Co., Ltd.(Tianjin, China). The filtrate was removed, rapidly cooled on ice to quench the reaction and analyzed by high-performance liquid chromatography (HPLC).

### 3.5. Kinetic Modeling of N-Acetylneuraminic Acid Degradation 

Data on the degradation of Neu5Ac were fitted against time to calculate the first-order degradation kinetics. The reaction rate constants (k) for Neu5Ac degradation at different temperatures were calculated via linear regression analysis using Equation (1) [26,27]: C = C_0_ exp (−k·t)(1)
where t is the time (h), C_0_ and C are the Neu5Ac concentrations (g/L) at time zero and time t, respectively, and k is the rate constant (min^−1^).

The decimal reduction time (D-value) and the half-life of degradation (t_1/2_) were calculated using Equations (2) and (3) [27,28,29], respectively.
(2)$$D\;=\;\frac{\ln\;(10)}{k}$$
(3)$$t\frac{1}{2}=\frac{\ln\;(2)}{k}$$

The thermal resistance coefficient (z-value) is the temperature needed to change the D-value by one log unit, and it was obtained by plotting the D-values on a log scale versus the corresponding temperatures.

It is possible to establish a relationship between the value of k and temperature by fitting the data to the Arrhenius equation [26,27]: (4)$$K\;=\;K_{0}\;\exp\;[\frac{-\mathrm{Ea}}{\mathrm{RT}}]$$

The enthalpy of activation (ΔH) and the free energy of inactivation (ΔG) at each temperature were obtained using Equations (5) and (6) [28,29], respectively:ΔH = Ea − RT(5)
(6)$$\Delta G\;=\;-\mathrm{RT}\;\ln[\frac{\mathrm{kh}}{k_{B}T}]$$
where h is the Planck’s constant (6.6262 × 10^−34^ J/s) and k_B_ is the Boltzmann’s constant (1.3806 × 10^−23^ J/K).

From Equations (5) and (6), it is possible to calculate the activation entropy (ΔS) [28,29]:(7)$$\Delta S\;=\;\frac{\left(\Delta H-\Delta G\right)}{T}$$

### 3.6. The Effect of H_2_O_2_ on the Stability of Neu5Ac 

Neu5Ac samples with pH 5.0, 7.0 and 9.0 were prepared as described above, after which 1.0, 2.0%, 3.0% or 4.0% H_2_O_2_ were added. The resulting reaction mixtures were kept at room temperature for 6 h prior to analysis.

### 3.7. Determination of Shelf Life

Neu5Ac solutions with pH values adjusted to 5.0, 7.0 and 9.0 were analyzed during storage for six weeks. The sample bottles were tightly covered with aluminum foil to avoid evaporation. Samples of the Neu5Ac solution were pipetted from each bottle every 2 weeks and filtered through a 0.22 µm pore-size membrane prior to HPLC analysis. The shelf life, which was defined as the time required to reach 90% of the labeled amount under defined storage conditions [16], was calculated using Equation (8).
(8)$$t_{90\%}=\;\frac{0.105}{K}$$

### 3.8. The Determination of Neu5Ac Content and Analytical Method Validation

The contents of Neu5Ac in the samples was determined by HPLC as reported before [30] with some modifications as follows: The analysis was performed on a Prominence LC-16 series HPLC system (Shimadzu, Kyoto, Japan) equipped with an SPD-16 ultraviolet detector. Samples were filtered through 0.22 µm pore-size membranes and analyzed on an Aminex HPX-87H column (300 mm × 7.8 mm × 9 µm; BioRad, Hercules, CA, USA) at 65 °C with 5 mM H_2_SO_4_ as the mobile phase with a flow rate of 0.6 mL/min. The injection volume was 20 µL. Neu5Ac was detected at 210 nm at an elution time of 8.5 min. 

Accuracy, precision, speciality, range, linear, limits of detection and limits of quantitation were used to verify the HPLC method validation. The recovery rate of the sample was between 99.3 and 100.4% (Appendix A). The RSD of instrument, method and method reproducibility were 0.15% (Appendix A), 0.08% (Appendix A) and 0.025% (Appendix A) respectively. The resolution was 3.532 (Appendix A), which proved that this method is with good specificity. The linear range of the method was 0.1–0.5 g/L (Appendix A). The test results of linear verification show a good linear relationship (R^2^ = 0.9999) (Appendix A). The limits of detection was 4.59 × 10^−6^ g/L and the limits of quantitation was 1.53 × 10^−5^ g/L (Appendix A). The verification results show that the analysis method was in accord with the purpose and requirements of the test.

### 3.9. Statistical Analysis

All experiments were done in triplicates and the results were expressed as mean values. The uncertainty of the experimental data was expressed as standard deviations from triplicate values, which were calculated using Microsoft Excel 2019 (Microsoft Corp., USA) and illustrated as error bars. SPSS 20.0 and one-way analysis of variance (ANOVA) were used for statistical analysis. *p* < 0.05 was considered to be statistically significant.

## 4. Conclusions

Neu5Ac showed good thermal stability at pH 3.0–10.0, especially at pH 7.0. The degradation rate of Neu5Ac was high in strong acids (pH 1.0 and pH 2.0) and strong alkali (pH 11.0 and pH 12.0). The thermal degradation of Neu5Ac followed first-order reaction kinetics at all investigated temperatures (60, 70, 80 and 90 °C) and pH values (1.0, 2.0, 11.0 and 12.0). The degradation under the influence of strong acid and strong alkali was nonspontaneous and endothermic. The process of Neu5Ac oxidation by H_2_O_2_ was consistent with the first-order kinetics at pH 5.0, 7.0 and 9.0. Although the degradation rate was the lowest at pH 9.0, the k constant was less affected by the pH. The addition of oxidants such as H_2_O_2_ to Neu5Ac preparations should be avoided, and it should be stored in an anoxic environment. This research provides kinetic data that can be used to predict product shelf lives at different temperatures and pH values. The degradation of Neu5Ac in aqueous solution followed first-order kinetics. The estimated shelf life of Neu5Ac at 25 °C in solutions with pH 5.0, 7.0 and 9.0 was 199, 817 and 387 days, respectively. These findings are useful as theoretical guidance for future research on the utilization of Neu5Ac.

## Figures and Tables

**Figure 1 molecules-25-05141-f001:**
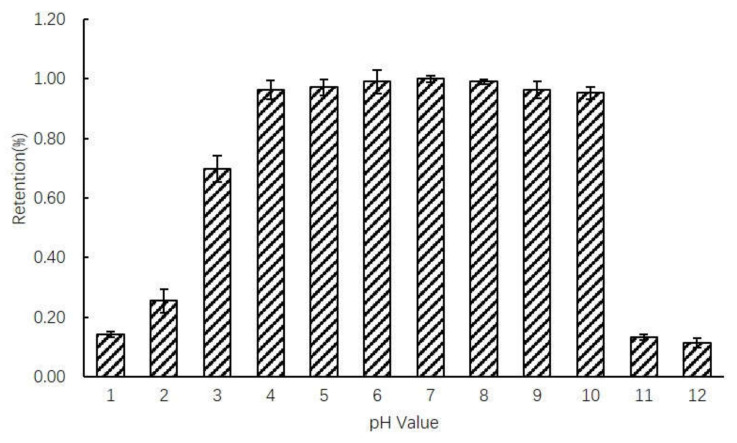
The impact of autoclave sterilization on the stability of Neu5Ac.

**Figure 2 molecules-25-05141-f002:**
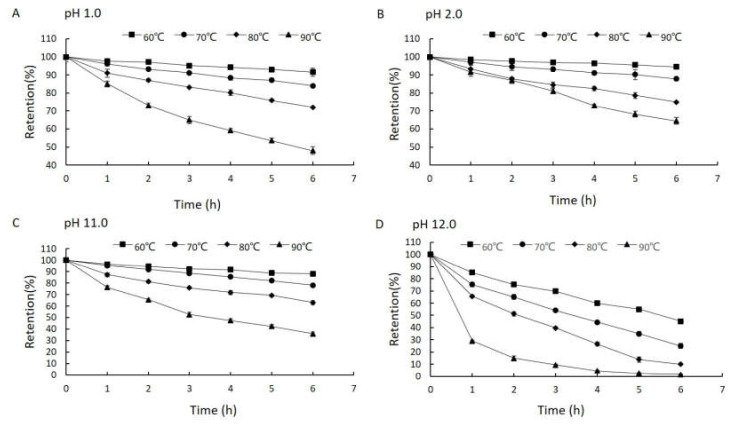
The residues of Neu5Ac solution after various heating temperatures, times and pH treatments.

**Figure 3 molecules-25-05141-f003:**
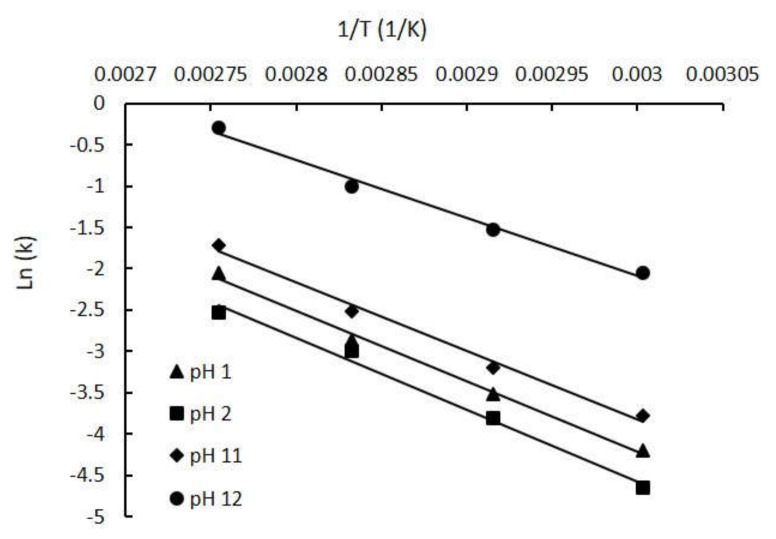
Arrhenius plots for Neu5Ac degradation in aqueous solutions at pH 1.0, 2.0, 11.0 and 12.0.

**Figure 4 molecules-25-05141-f004:**
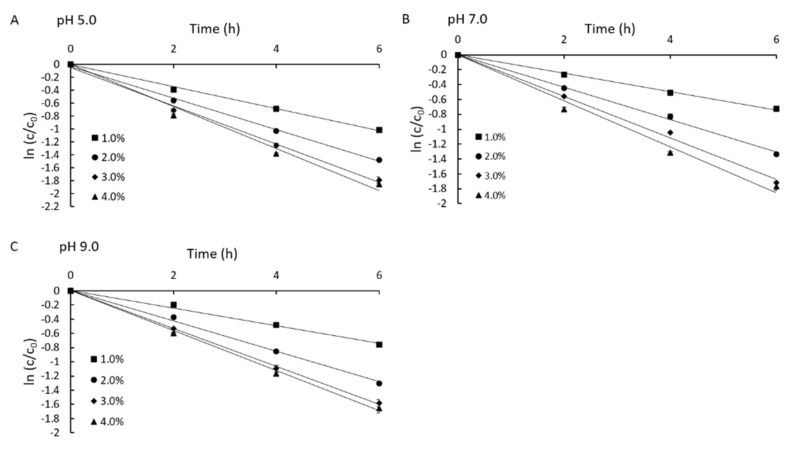
The effect of H_2_O_2_ on the stability of Neu5Ac.

**Figure 5 molecules-25-05141-f005:**
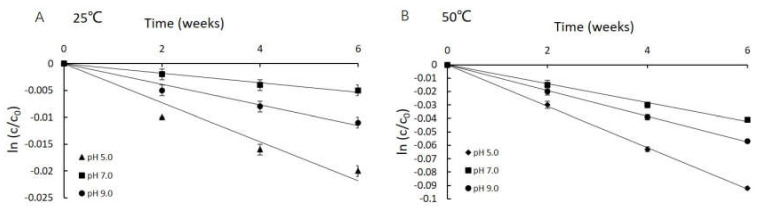
Plots of −ln (C_t_/C_0_) versus time for samples stored at different pH values and temperatures of 25 °C and 50 °C.

**Table 1 molecules-25-05141-t001:** Kinetic parameters of Neu5Ac at pH 1.0, 2.0, 11.0 and 12.0.

pH	T (°C)	K (h^−1^)	R^2^	t_1/2_ (h)	Ea (kJ/mol)	D (h)	Z (°C)	ΔH (kJ/mol)	ΔG (kJ/mol)	ΔS (J/mol)
1.0	60	0.0150 ^d^	0.9797	46.21	71.25	153.51	32.3	68.48	93.49	−75.10
	70	0.0296 ^c^	0.9673	23.42		77.79		68.40	94.44	−75.93
	80	0.0567 ^b^	0.9838	12.22		40.61		68.32	95.37	−76.66
	90	0.1288 ^a^	0.9763	5.38		17.88		68.23	95.68	−75.62
2.0	60	0.0095 ^d^	0.9838	72.96	72.50	242.38	31.5	69.73	94.75	−75.12
	70	0.0222 ^c^	0.9668	31.22		103.72		69.65	95.26	−74.66
	80	0.0498 ^b^	0.9728	13.92		46.24		69.57	95.75	−74.19
	90	0.0799 ^a^	0.9693	8.68		28.82		69.48	97.12	−76.14
11.0	60	0.0230 ^d^	0.9626	30.14	68.84	100.11	33.3	66.07	92.30	−78.78
	70	0.0406 ^c^	0.9422	17.07		56.71		65.99	93.54	−80.33
	80	0.0803 ^b^	0.9959	8.63		28.67		65.91	94.35	−80.58
	90	0.1808 ^a^	0.9535	3.83		12.74		65.82	94.66	−79.44
12.0	60	0.1277 ^d^	0.9596	5.43	58.50	18.03	39.2	55.74	87.56	−95.55
	70	0.2177 ^c^	0.9790	3.18		10.58		55.66	88.75	−96.48
	80	0.3684 ^b^	0.9871	1.88		6.25		55.58	89.88	−97.18
	90	0.7493 ^a^	0.9883	0.93		3.07		55.49	90.37	−96.07

^1^ a–d Values with the different letter in a column are significantly different (*p* > 0.05).

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
