# Peer review of "Degradation Kinetics and Shelf Life of N-acetylneuraminic Acid at Different pH Values"

_molecules, 2020, doi:10.3390/molecules25215141_

Round 1

Reviewer 1 Report

The paper:

Degradation Kinetics and Shelf Life of

N-acetylneuraminic Acid at Different pH Values

The paper presents the kinetic study of the degradation of N-acetylneuraminic acid at several pH values. It is a common work, in my opinion without any fabulous achievement. Quite simple for the requirements of Molecules magazine. For a clearer understanding I recommend moving the materials and methods chapter before results and discussions at least for kinetics studies.

Reviewer 2 Report

The purpose of the article “Degradation Kinetics and Shelf Life of
3 N-acetylneuraminic Acid at Different pH Values” was to analyze the influence of pH and temperature on the stability of Neu5Ac in aqueous solution.

The article is easy to understand and presents the proposed objectives, but I have one requests to make:

1 - Improve the discussion of results and detail the applications that can benefit from this study.

Reviewer 3 Report

Manuscript Number: Molecules-953617

This work reports the degradation kinetics and shelf life of Neu5AC under different storage conditions.  The methodology already known and well documented in the literature, but there is neither contribution to the knowledge nor novel facts with respect to this. However, the obtained data could be useful for further research with this natural monosaccharaide. The authors have used a HPLC method to determine the concentration of drug, but they have only described the experimental conditions (see lines 179 and 187), but any information about the validation and quantitation of drug was reported. This type of information is vital in the stability study.  

Kinetic modelling should be justified. In my opinion, it is necessary to support the applied model by using appropriate statistical analysis.  

For general case, long term, accelerated and intermediate (where appropriate) storage conditions are detailed at the ICH Q1A-(R2) guideline: Stability Testing of new drugs substances and products. For example, long term should conducted at 25±2°C during 12 months, whereas the accelerated study at 40±2°C during 6 months. However, the authors performed the long-term study during six weeks and accelerated conditions at 50°C. Why did they use these conditions? This should explained.

Other minor comments

  • Line 203, equation (4). There is a mistake, lack the minus sign in the Energy Activation term. Please, revise                                                              
  • The reference #27 is incomplete. The year not provided.

In this Reviewer’s opinion, the manuscript is not suitable for publication in Molecules Journal in actual form.

Round 2

Reviewer 1 Report

In my opinion the paper can be published in this form. The recommendations were taken into account.

Author Response

We appreciate for your warm work earnestly. Thank you very much for your comments.

Reviewer 3 Report

This manuscript is a revised version. 

The authors have provided an appropriate explanation about the different questions from other reviewers, and I am in agreement with the author’s reply.

The  analytical method validation is not total, the accuracy and precision were only analyzed, but this information is not include in the revised version, Why? 

In addition, the quality and interest of the paper have improved.  In this Reviewer’s opinion, the manuscript is suitable for publication in the Molecules Journal in present form.
